# An inventory of human night-biting mosquitoes and their bionomics in Sumba, Indonesia

Lepa Syahrani[1]ᵒ, Dendi H. Permana[1]ᵒ, Din Syafruddin[1,2]*, Siti Zubaidah[1]ᵒ, Puji B. S. Asih[1], Ismail E. Rozi[1], Anggi P. N. Hidayati[1], Sully Kosasih[1], Farahana K. Dewayanti[1], Nia Rachmawati[1], Rifqi Risandi[1], Michael J. Bangs[3,4†], Claus Bøgh[5], Jenna Davidson[6], Allison Hendershot[6], Timothy Burton[6], John P. Grieco[6], Nicole L. Achee[6], Neil F. Lobo[6]

1 Eijkman Institute for Molecular Biology, Jakarta, Indonesia, 2 Department of Parasitology, Faculty of Medicine, Hasanuddin University, Makassar, Indonesia, 3 Public Health and Malaria Control, PT Freeport Indonesia, International SOS, Kuala Kencana, Papua, Indonesia, 4 Department of Entomology, Faculty of Agriculture, Kasetsart University, Bangkok, Thailand, 5 The Sumba Foundation, Public Health and Malaria Control, Bali, Indonesia, 6 Department of Biological Sciences, Eck Institute for Global Health, University of Notre Dame, Indiana, United States of America

ᵒ These authors contributed equally to this work.
† Deceased.
* dinkarim@yahoo.com

**Data Availability Statement:** All relevant data are within the manuscript.

**Funding:** This study was funded by an award from the Bill and Melinda Gates Foundation (BMGF) to

## Abstract

Mosquitoes are important vectors that transmit pathogens to human and other vertebrates. Each mosquito species has specific ecological requirements and bionomic traits that impact human exposure to mosquito bites, and hence disease transmission and vector control. A study of human biting mosquitoes and their bionomic characteristics was conducted in West Sumba and Southwest Sumba Districts, Nusa Tenggara Timur Province, Indonesia from May 2015 to April 2018. Biweekly human landing catches (HLC) of night biting mosquitoes both indoors and outdoors caught a total of 73,507 mosquito specimens (59.7% non-*Anopheles*, 40.3% *Anopheles*). A minimum of 22 Culicinae species belonging to four genera (*Aedes*, *Armigeres*, *Culex*, *Mansonia*), and 13 Anophelinae species were identified. *Culex quinquefasciatus* was the dominant Culicinae species, *Anopheles aconitus* was the principal *Anopheles* species inland, while *An. sundaicus* was dominant closer to the coast. The overall human biting rate (HBR) was 10.548 bites per person per night (bpn) indoors and 10.551 bpn outdoors. Mosquitoes biting rates were slightly higher indoors for all genera with the exception of *Anopheles*, where biting rates were slightly higher outdoors. Diurnal and crepuscular *Aedes* and *Armigeres* demonstrated declining biting rates throughout the night while *Culex* and *Anopheles* biting rates peaked before midnight and then declined. Both anopheline and non-anopheline populations did not have a significant association with temperature ($p = 0.3$ and $0.88$ respectively), or rainfall ($p = 0.13$ and $0.57$ respectively). The point distribution of HBR and seasonal variables did not have a linear correlation. Data demonstrated similar mosquito–human interactions occurring outdoors and indoors and during early parts of the night implying both indoor and outdoor disease transmission potential in the area–pointing to the need for interventions in both spaces. Integrated vector analysis

the University of Notre Dame (Grant# OPP1081737) granted to NLA and NFL, the Government of Indonesia granted to DS, Ministry of Research and Technology/National Research and Innovation Agency through Eijkman Institute for Molecular Biology. The funders had no role in study design, data collection, and analysis, decision to publish, or preparation of the manuscript.

**Competing interests:** The authors have declared that no competing interest exist. Author Michael J. Bangs was unable to confirm their authorship contributions. On their behalf, the corresponding author has reported their contributions to the best of their knowledge.

frameworks may enable better surveillance, monitoring and evaluation strategies for multiple diseases.

## Author summary

This study outlines the array of mosquitoes that bite humans at night on the island of Sumba, Indonesia, with data on behavioural traits that impact when and where disease transmission may occur. Biweekly human landing catches (HLCs) were performed in four selected houses in 12 clusters (villages) from sunset to sunrise over a three years period (May 2015 to April 2018). The collection and analysis of 73,507 mosquito specimens revealed the presence of various species of *Anopheles*, *Aedes*, *Culex*, *Armigeres* and *Mansonia*, that potentially transmit several diseases including malaria, filaria, dengue and other mosquito borne viral diseases. Even though these data represent only night-time collections, this represents a comprehensive geographic description and inventory of species, bionomics and temporal distribution of mosquitoes on the island of Sumba. Data demonstrate that the high diversity of species with associated diversity in behaviours results in mosquito-human contact occurring throughout the night and both indoors and outdoors–relevant to both disease transmission and intervention applicability. Vector specific behaviours are specifically relevant to intervention strategies for specific diseases. The use of molecular methods to determine and validate morphological identification of specimens resulted in the characterization of multiple novel sequences–indicating the presence of undescribed species, members of cryptic species complexes or species without molecular data. Species identification using molecular methods are essential towards determine vector species compositions–especially in areas where data is absent. Though the correlation between temperature, rainfall and HBR was not statistically significant, the presence of mosquito populations throughout the year allow for perennial transmission of mosquito-borne diseases. Overall, these findings represent baseline and novel data for Sumba and may be utilized to develop disease and vector-specific or integrated strategies that mitigate the transmission of mosquito borne diseases in Indonesia.

## Introduction

Mosquitoes (Order: Diptera; Family: Culicidae) are an important group of arthropods that transmit diseases to humans and animals through their blood feeding behaviour [1]. Diseases transmitted by mosquitoes include malaria, dengue, Zika, filariasis, Japanese encephalitis, and chikungunya, all documented in Indonesia, contributing to mortality and morbidity throughout the country with millions of people at risk of infection [1,2].

Globally, there are approximately 3,200 identified mosquito species in three subfamilies: Toxorhynchitinae (*Toxorhynchites*), Culicinae (*Aedes*, *Culex*, *Mansonia*, *Armigeres*) and Anophelinae (*Anopheles*) [3]. Of the many species included in these subfamilies, only a subset have been confirmed as vectors of diseases. Transmitted by *Anopheles*, malaria is endemic in Sumba Island with higher clinical cases in the rainy season compared to the dry season—attributed to increased mosquito vector populations associated with rain [4]. Kodi Balaghar sub district in southwest Sumba district is also endemic for filariasis [5]. Filariasis is caused by filarial nematode worms *Wucheria bancrofti*, *Brugia malayi* and *B. timori* and are transmitted by mosquito species within multiple genera including *Mansonia*, *Anopheles*, *Culex* and *Aedes* [6]. *Mansonia*

*uniformis* and *An. nigerrimus* have been confirmed as *B. malayi* vectors while *An. barbirostris* has been reported as the vector of *B. timori*, commonly found in East Nusa Tenggara and South Maluku regions in Indonesia. *Anopheles sundaicus*, *An.vagus* and *An. subpictus* are vectors of *W. bancrofti* in East Nusa Tenggara [7,8]. Dengue transmitted by *Aedes* species is also prevalent in East Nusa Tenggara Province with incidence rates reported in 2017 as high as 19.5 per 100,000 people [9]. Primary dengue vectors include *Ae. albopictus* and *Ae. aegypti*, with the Chikungunya virus also being transmitted by *Ae. aegypti* [10,11].

Interventions directed at mosquito vectors of disease rely on vector behavioural traits. Efficient and impactful intervention strategies are dependent on vector knowledge that describe their behaviour and ecology in combination with the epidemiology of the disease in humans. Species specific bionomic traits and species specific drivers of transmission, rely on morphological and molecular characterization of the vector species. Misidentification of vectors impacts potential downstream analyses and intervention strategies. This effect was seen in central Vietnam, where *An. varuna* was mistakenly described as *An. minimus* as a primary vector. Vector control efforts were hence directed towards described *An. minimus* population peaks resulting in wasted resources since local *An. varuna* is highly zoophagic and unlikely to be a malaria vector [12]. The correct identification of any mosquito implicated as a vector is key to successful control or elimination measures. Similarly the comprehension of species specific impacts of interventions allowed for the description of gaps in protection in Kenya [13] and the Solomon Islands [14].

Standard practices towards the identification of mosquito species include morphological and sometimes, molecular identification, combined with parallel ecological and bionomic data used to improve the accuracy of species identification [15]. Morphological identification using regional morphological keys is commonly used as it is less labour intensive and time efficient. However, the specificity and sensitivity of morphological identification may be compromised based on local applicability of the specific morphological identification keys, appropriate training and human error, as well as difficulty associated with identifying sibling or cryptic species, regional morphological variants, and new or novel species. Thus, molecular identification may be used in conjunction with morphological methods along with ecological analysis towards improving accuracy and produce more informative data.

Towards filling this important knowledge gap in West Sumba, Indonesia, this study aimed to catalogue and identify human, night-biting mosquito species, their temporal presence and bionomic characteristics.

## Methods

### Ethics statement

Ethical review and approval was granted by the Ethics Committee (EC) of the Faculty of Medicine, Universitas Hasanuddin, Indonesia and the University of Notre Dame, USA. Verbal and written informed consent was obtained from local volunteers for landing catches, who were all more than 18 years old and from house owners.

### Study site and design

This study was conducted in Southwest and West Sumba Districts, East Nusa Tenggara Province located on the island of Sumba in the eastern part of Indonesia (Fig 1) from May 2015 to April 2018. The climate is tropical, with a dry season from May to November and a wet season from December to April. This dataset was collected as part of a parent, cluster randomized, double-blind, placebo-controlled, clinical trial, that measured the public health impact of a spatial repellent on malaria incidence [4]. Of the 24 clusters included in the parent study, 12

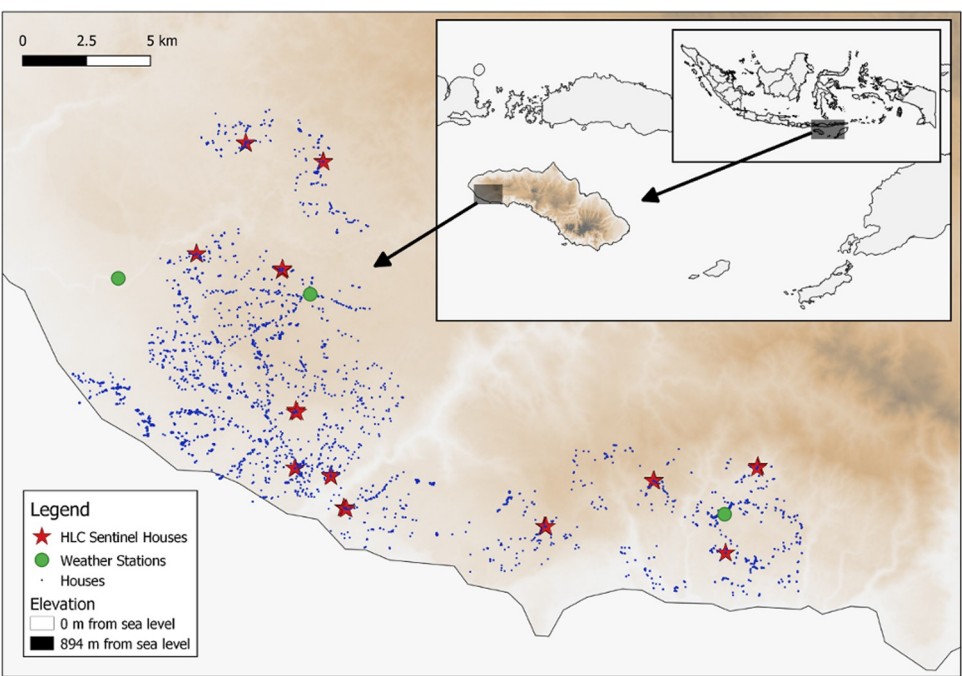

**Fig 1. Map of the study site: 4 sentinel houses in each clusters are marked by stars.** A map depicting the location of study site in Southwest and West Sumba Districts, East Nusa Tenggara Province, Indonesia (http://www.naturalearthdata.com/).

geographically distributed clusters were utilized for entomological follow-up with bi-weekly human landing catches (HLCs). Climatic parameters such as temperature and rainfall were obtained from HOBO weather stations installed in three locations across the study area. Monthly average rainfall was calculated from the daily values of the three weather stations. Rainfall data was obtained from July 2016 to April 2018. Temperature data was obtained from May 2015 to April 2018.

## Mosquitoes collection

In each cluster, four traditional houses were selected having similar size and design, and with house owners agreeing to participate in the study. These sentinel houses were located close to permanent or semi-permanent *Anopheles* larval sites. Indoor and outdoor (household veranda) paired HLCs were performed by a two-member team (local volunteers with informed consent) per house, in four sentinel HLC houses in each cluster [4]. Host seeking mosquitoes landing on exposed feet and legs were caught using an aspirator for 50 min each hour from 1800 h to 0600h. Mosquitoes were held in individual paper cups labelled for each hour, location (indoor or outdoor) and house code. Female mosquito specimens were transported to an on-site study laboratory for further processing. In total, with 52 nights of HLCs in 12 clusters, and in 4 houses each (inside and outside) there were 4,992 person nights of HLC collections (2,496 indoors and outdoors each).

## Mosquitoes identification

Mosquito specimens were morphologically identified to genera (*Culex*, *Anopheles*, *Aedes*, *Armigeres*, and 'other') using taxonomic keys [16,17]. Samples were stratified by morphological species, cluster, location, and time of capture, and a random subset (representing at least

10% of each morphologically identified species) spanning all strata were sequenced at the internal transcribed spacer2 (ITS2) and/or cytochrome c oxidase subunit I (COI) loci towards species determination [18–24]. A larger subset of *Anopheles* was analysed based on the parent study focus on *Anopheles* and malaria.

For molecular species confirmation, DNA was extracted from whole specimens using a Chelex-100 ion exchange (BioRad Laboratories, Hercules, CA, USA). PCR amplicons were sequenced at the Eijkman Institute for Molecular Biology, Indonesia, and the University of Notre Dame, USA. Comparisons were made between morphological and molecular species identifications. Final species confirmation required high sequence identity (thresholds of 97% for ITS2 and 94% for CO1) to sequences in multiple databases. CO1 and ITS2 database comparisons for each sample were paired to determine species when either CO1 or ITS2 alone did not produce significant results to voucher sequences [18,21,23]. Consensus sequences were manually inspected for insertions, deletions, and repeat regions to ensure these sequence differences did not inflate divergence and decrease identity scores.

## Analysis

The night-time human biting rate (HBR) was determined as bites per person per night (bpn) or bites per person per hour (bph). The relationship between independent variables (climatic parameters) and dependent variables (HBR) were tested using the Pearson correlation coefficients and scatterplot correlation. Tests were also conducted to determine the relationship between mosquito abundance and rainfall and temperature [25]. Bionomic inferences were made at the genus level for *Anopheles*, *Armigeres*, *Aedes* and *Culex*, as well as for the five most common *Anopheles* species. Consensus sequences of each sequence group were compared (BLASTn) to the NCBI nr and BOLD [26] databases to identify species. Species belonging to the *An. barbirostris* species complex were compared to type specimen sequences [27,28] towards molecular identification.

## Results

### Mosquitoes composition

During the 52 nights of human landing collections, a total of 73,507 adult female Culicidae mosquitoes were captured. These samples included *Culex* (40.29%; n = 29,612), *Anopheles* (40.30%; n = 29,636); *Aedes* (12.86%, n = 9,451), and *Armigeres* (6.27%; n = 4,608). Unidentified female mosquitoes and *Mansonia* species were grouped together as 'others' (0.27%; n = 200). Overall, 50.0% (36,759) of the mosquitoes were captured outdoors and 50.0% (36,748) indoors.

Average daily human biting rates (HBR) for the 35 months of the study were determined with the assumption that landing rates in the HLCs were a proxy for HBRs (Fig 2). Indoor and outdoor biting rates did not vary significantly within for *Aedes*, *Armigeres*, or *Culex*, with slightly higher indoor biting seen. Of the five dominant *Anopheles* species, four had slightly higher outdoor biting rates (*An. aconitus*: 1.62 bpn indoor versus 1.92 bpn outdoor; *An. flavirostris*: 1.14 bpn indoor versus 1.46 bpn outdoor; *An. sundaicus*: 0.92 bpn indoor versus 1.10 bpn outdoor; *An. vagus*: 1.32 bpn indoor versus 1.36 bpn outdoor) while *An. tesselatus* had a slightly higher indoor biting rate (1.11 bpn indoor versus 1.07 bpn outdoor) (Fig 3). Biting rates varied by geography of collection and was usually associated with the availability of larval sites and with agricultural irrigation. Interestingly, the biting rate of *An. sundaicus*, the historical primary vector in the area, dropped from 8.79 bpn (June 2015-December 2015; with a high of 24.81 bpn in August 2015) to an average of 0.29 bpn post December, 2016. The biting

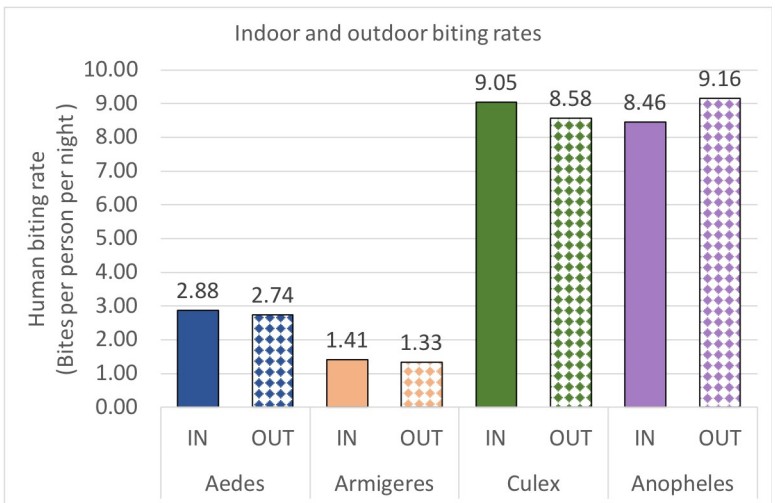

**Fig 2. Indoor and outdoor biting rates of Culicidae.**

densities of *An. sundaicus* only recovered slightly between August and October 2017 (a high of 4.14 bpn in September, 2017).

All mosquitos were found to bite throughout the night both indoors and outdoors. *Aedes* and *Armigeres* species tended to have their biting rates drop over the course of a night–*Aedes*: from 0.37 bites per person per hour (bph) between 1800 to 1900h, to 0.16 bph between 0500 to 0600h; *Armigeres*: from 0.23 bph between 1800h to 1900h, to 0.16 bpn between 0500h to 0600h). *Culex* biting rates peaked between 1900 h and 2100 h (0.87 bph) while *Anopheles* bites peaked between 2100h and 2300h (0.85 bpn). Both *Culex* and *Anopheles* biting rates dropped to 0.6 and 0.59 bph respectively at the 0500 to 0600h collection timepoint (Fig 4).

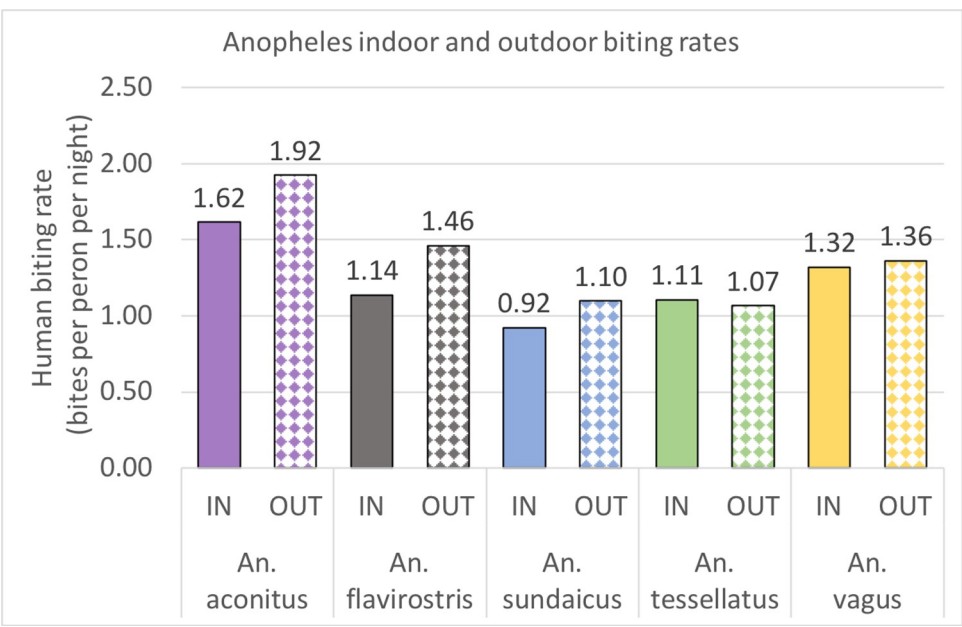

**Fig 3. Indoor and outdoor biting rates of *Anopheles* species.**

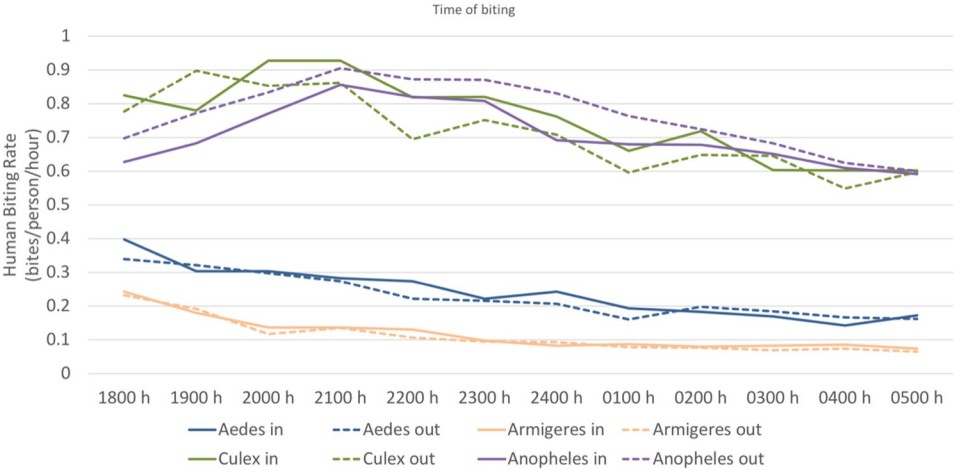

**Fig 4. Indoor and outdoor biting rates of *Aedes*, *Armigeres*, *Culex* and *Anopheles* species over the course of a night.**

The five dominant *Anopheles* species had similar biting rates indoors and outdoors over the course of the night. Though HBRs did not vary significantly over the night between species, *An. aconitus* and *An. tesselatus* tended to have slightly higher hourly biting rates from 1800 to 2200h (0.17 bph), both *An. flavirostris* and *An. sundaicus* had peak biting between 2100 and 0100h (01.3 bph), while *An. vagus* had a fairly consistent HBR over the night, with a slightly higher biting rate from 0200 to 0600h (0.12 bph).

## Morphological species identifications

Morphological species identification was performed on *Aedes* (n = 589), *Armigeres* (n = 183), *Culex* (n = 1,312), *Mansonia* (n = 44) and *Anopheles* (n = 29,636) specimens. Morphologically identified *Culex* species included *Cx. quinquefasciatus* (the most abundant species), followed by *Cx. gelidus*, *Cx. vishnui*, *Cx. tritaeniorhynchus*, *Cx. fuscocephala*, *Cx. bitaeniorhynchus*, *Cx. sinensis*, *Cx. halifaxii*, and *Cx. whitmorei*. *Aedes* species morphologically identified included *Ae. aegypti*, *Ae. albopictus*, *Ae. poicilius*, *Ae. vexans*, and *Ae. (mucidus)* spp. Morphologically identified *Armigeres* species included *Armigeres malayi* and *Ar. subalbatus*, while *Mansonia uniformis* was identified in the 'other' set.

All *Anopheles* specimens were morphologically identified to species. Of the 13 *Anopheles* species identified, the five dominant species included *An. aconitus* (20.07% of *Anopheles*, n = 5,946), *An. vagus* (15.2% of *Anopheles*; n = 4,502); *An. flavirostris* (14.74% of *Anopheles*; n = 4,365), *An. tesselatus* (12.32% of *Anopheles*; n = 3,651), and *An. sundaicus* (11.45% of *Anopheles*; n = 3,392) (Table 1). The remaining eight identified species of *Anopheles* representing 26.22% of the total *Anopheles* comprised of *An. annularis*, *An. balabacensis*, *An. barbirostris*, *An. indefinitus*, *An. kochi*, *An. leucosphyrus*, *An. maculatus*, and *An. subpictus*. *Anopheles aconitus* was the predominant species found in the upland interior locations whereas *An. sundaicus* was dominant in coastal areas. A small number (n = 20) of *Anopheles* specimens remain unidentified morphologically.

## Molecular species identifications

There were seven *Aedes* taxa identified molecularly, two taxa were identified to species while five remained unidentified. Identified species had similarities higher than the thresholds (97%

**Table 1. Species identifications.** Molecular species identifications (with the number of specimens) are listed with ITS2 and CO1 similarities to sequences in the databases (NCBI and BOLD). Morphological identifications of the molecularly identified specimens are also listed.

| Genera | Molecular species (#) | Sequence Type (% Identity) | | Morphological Identifications (#) |
|---|---|---|---|---|
| | | ITS2 | CO1 | |
| Culex | Cx. bitaeniorhynchus (5) | 97.9 | 100 | Cx. vishnui (4), Cx. sinensis (1) |
| | Cx. fuscochepala (5) | - | 100 | Cx. tritaeniorhynchus (4), Cx. quinquefasciatus (1) |
| | Cx. gelidus (37) | 99.7 | 100 | Cx. gelidus (33), Cx. tritaeniorhynchus (2), Cx. vishnui (2) |
| | Cx. nigropunctatus (4) | - | 98.7 | Cx. quinquifasciatus (4) |
| | Cx. orientalis (2) | 98.2 | - | Cx. vishnui (1), Cx. sinensis (1) |
| | Cx. pseudovishnui (5) | 98.5 | - | Cx. tritaeniorhynchus (2), Cx. vishnui (2), Ae. vexans (1) |
| | Cx. quinquefasciatus (4) | 99.2 | - | Cx. quinquefasciatus (4) |
| | Cx.tritaeniorhynchus (17) | 99.4 | 99.6 | Cx. tritaeniorhynchus (10), Cx. vishnui (6), Cx. quinquefasciatus (1) |
| | Cx. vishnui (6) | 99.3 | - | Cx. vishnui (3), Cx. tritaeniorhynchus (3) |
| | Culex species 1 (3) | 86 | - | Cx. tritaeniorhynchus (1), Cx. vishnui (1), Cx. quinquefasciatus (1) |
| | Culex species 2 (4) | 78.6 | - | Cx. quinquefasciatus (4) |
| Aedes | Ae. albopictus (12) | 99.2 | 99.9 | Ae. albopictus (11) |
| | | | 99.4 | Ae. albopictus (1) |
| | Ae. vexans (7) | 93.7 | 99.9 | Ae. vexans (5), Cx. vishnui (2) |
| | Aedes species 1 (5) | 81.5 | - | Ar. subalbatus (5) |
| | Aedes species 2 (5) | 80 | - | Ar. malayi (2), Ar. subalbatus (2), Cx. tritaeniorhynchus (1) |
| | Aedes species 3 (36) | 89.3 | 93.4 | Ae. albopictus (25), Ae. vexans (11) |
| | Aedes species 4 (4) | 94.4 | - | Ae. albopictus (4) |
| | Aedes species 5 (1) | - | 89 | Ae. poicilius (1) |
| Armigeres | Ar. malayi (16) | - | 100 | Ar. malayi (14), Ar. subalbatus (2) |
| | Ar. subalbatus (27) | 94.3 | 97.3 | Ar. subalbatus (27) |
| | Ar. cf. subalbatus (3) | | 91.1 | Ar. subalbatus (2), Cx. quinquefasciatus (1) |
| Other | Mansonia uniformis (12) | 100 | - | Ae. poicilius (12) |
| Anopheles | An. aconitus (1,822) | 100 | 99 | An. aconitus (1,644), An. annularis (9), An. flavirostris (134), An. barbirostris (3), An. farauti (1) An. kochi (6), An. maculatus (10), An. tesselatus (6), An. vagus (5), An. sundaicus (1), An. leucosphyrus (1), An. indefinitus (1), An. karwari (1) |
| | An. annularis (491) | 99.8 | 95.81 | An. annularis (459), An. aconitus (12), An. barbirostris (3), An. subpictus (1), An. flavirostris (1), An. kochi (1), An. maculatus (2), An. sinensis (2), An. sundaicus (1), An. tesselatus (3), An. vagus (6) |
| | An. balabacensis (34) | 99.9 | - | An. leucosphyrus (30), An. flavirostris (1), An. maculatus (3) |
| | An. barbirostris clade 1 (116) | 99.5 | 100 | An. barbirostris |
| | An. barbirostris clade 2 (74) | 99.5 | - | An. barbirostris |
| | An. sundaicus s.l (296) | 100 | 100 | An. sundaicus (258), An. aconitus (10), An. barbirostris (17), An. flavirostris (5), An. vagus (3), An. annularis (2), An. subpictus (1) |
| | An. flavirostris (1,097) | 100 | 98 | An. flavirostris (955), An. aconitus (102), An. annularis (5), An. barbirostis (7), An. maculatus (6), An. sundaicus (10), An. kochi (3), An. leucosphyrus (2), An. montanus (1), An. subpictus (2), An. tesselatus (2), An. vagus (2) |
| | An. indefinitus (3) | 100 | - | An. aconitus (1), An. sundaicus (2) |
| | An. kochi (758) | 100 | 100 | An. kochi (700), An. aconitus (6), An. barbirostris (9), An. tesselatus (16), An. vagus (12), An. annularis (3), An. flavirostris (3), An. maculatus (2), An. punctulatus (1), An. sinensis (1), An. subpictus (3), An. sundaicus (2) |
| | An. maculatus (335) | 100 | 100 | An. maculatus (309), An. aconitus (10), An. annularis (2), An. vagus (3), An. barbirostris (2), An. flavirostris (5), An. kochi (1), An. sundaicus (1), An. tesselatus (2) |
| | An. subpictus (68) | 99.8 | 94.39 | An. subpictus (31), An. indefinitus (8), An. vagus (23), An. sundaicus (2), An. maculatus (1), An. aconitus (1), An. tesselatus (2) |
| | An. tesselatus (876) | 100 | 99.84 | An.tesselatus (799), An. annularis (4), An. kochi (7), An. maculatus (8), An. aconitus (7), An. barbirostris (6), An. barbumrosus (3), An. indefinitus (2), An. kochi (12), An. parangensis (1), An. sinensis (1), An. subpictus (4), An. flavirostris (7), An. vagus (15) |
| | An. vagus (1,004) | 98.7 | 100 | An. vagus (924), An. annularis (5), An. maculatus (5), An. subpictus (11), An. sundaicus (9), An. kochi (7), An. tesselatus (17), An. indefinitus (6), An. aconitus (7), An. annularis (5), An. barbirostris (2), An. flavirostris (14). |

for ITS2, and 94% for CO1) to sequences in the database (NCBI and BOLD) [29–31]. Identified species include *Ae. albopictus* and *Ae. vexans*. Two groups of *Ae. albopictus* CO1 sequences were detected that were 5% different from each other. Unidentified *Aedes* specimens had

sequences with low similarity (below 94%) to *Ae. aegypti*, *Ae. ochraceus*, and *Ae. geniculatus* (Table 1).

Of the three *Armigeres* species documented molecularly, *Ar. malayi* and *Ar. subalbatus* were identified to species, while the third (*Ar. cf. subalbatus*) set of sequences were only 91.12% similar to *Ar. subalbatus* CO1 sequences and consequently do not have a species designation due to the identity thresholds applied (Table 1).

Similarly, there were 11 taxa of *Culex* identified from ITS2 and/or CO1 sequences—including nine known and two unidentified species. The known species, with high sequence similarity in the databases, included *Cx. gelidus*, *Cx. quinquefasciatus*, *Cx. vishnui*, *Cx. tritaeniorhynchus*, *Cx. pseudovishnui*, *Cx. bitaeniorhynchus*, *Cx. orientalis*, *Cx. nigropunctatus*, and *Cx. fuscochepala*. The unidentified species, *Culex* species 1 and 2, were closest to *Cx. dolosus* (86.0%) and *Cx. palpalis* (78.6%)—similarity below conservative thresholds to confirm identity (Table 1). There were 13 molecularly identified *Anopheles* species including *An. aconitus*, *An. annularis*, *An. sundaicus s.l*, *An. balabacensis*, *An. barbirostris clade 1 (An. barbirostris s.s.)*, *An. barbirostris clade 2 (An. vanderwulpi)*, *An. flavirostris*, *An. indefinitus*, *An. kochi*, *An. maculatus*, *An. tesselatus*, *An. subpictus* and *An. vagus* (Table 1). *Anopheles barbirostris s.s.* and *An. vanderwulpi* were identified to species based on SNPs and homology to type sequences [28].

*Mansonia uniformis* was identified in the 'other' group, in addition to a single unidentified set of sequences pointing to a Dipteran species that remains unidentified. This unknown Dipteran had low identity (15%) to Culicidae sequences and was left out of the analysis.

## Bionomics—Seasonality

Weather stations recorded rainfall throughout the study period. Rainfall was analysed as a driver of seasonal mosquitoes density. The highest daily rainfall occurred in February 2017 (a mean of 24.73 mm), while the lowest was in August 2017 (mean of 0.19 mm). Overall, there were an increasing number of Culicidae specimens caught during the rainy season as compared to the dry season—with the most significant increase being in *Culex* and *Anopheles* species (Fig 5). Statistical analysis with a Pearson correlation test demonstrates that rainfall intensity did not have a significant correlation with the number of mosquitoes ($p = 0.3$ for anopheline, $p = 0.88$ for non- anopheline).

Temperature ranged between lows of 22.6°C and highs of 32.8°C without any significant short-term fluctuations. Statistical analyses (Pearson) demonstrated the lack a relationship between temperature and mosquito density ($p = 0.13$ for anopheline, $p = 0.57$ for non-

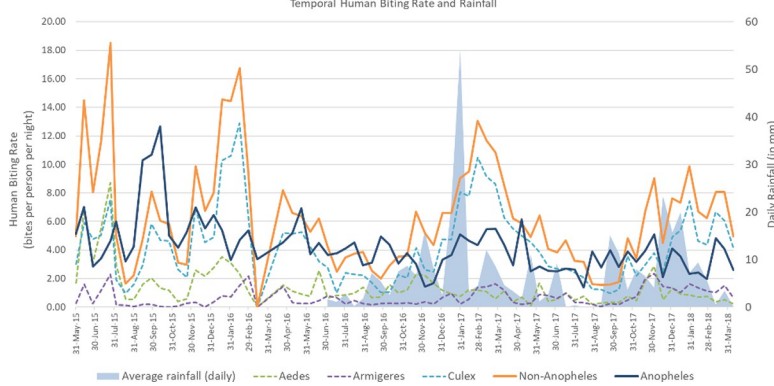

**Fig 5. Seasonality of Culicidae mosquito genera related to rainfall over the 35 months of data collection.**

anopheline). Similar results with scatterplot correlations, demonstrated that both anopheline and non-anopheline do not have linear correlations with rainfall or temperature.

## Discussion

The parent intervention trial enabled standardized, 35 month collections of night-time human host seeking mosquitos in Sumba, Indonesia. The use of sentinel structures with paired indoor and outdoor HLCs enabled the first time evaluation of species compositions, seasonal variations, population drivers, and human host-seeking behaviours of *Aedes*, *Armigeres*, *Culex* and *Anopheles* mosquitoes over the night, in Indonesia.

Approximately 73,507 mosquito specimens collected were found to consist of at least 35 molecularly determined species. Culicinae mosquitoes were separated and processed differently than *Anopheles*, as the parent study was directed at *Anopheles* and malaria. A smaller set of Culicinae mosquitoes were identified morphologically and molecularly (*Aedes*: 589 morphologically (6.23%), 126 molecularly (1.33%); *Armigeres*: 183 morphologically (3.97%), 45 molecularly (0.98%); *Culex*: 1,312 morphologically (4.43%), 118 molecularly (0.40%); and 'Other': 44 morphologically (22.00%), 14 molecularly (7%)). Though the samples were randomly selected to represent all collection times and sites, the lower morphological and molecular identifications for Culicinae mosquitoes indicates that some species may have been missed. The high stringency used for identifying species molecularly [18,21,23] enabled a conservative approach. Several Culicinae species remain unidentified primarily due to a lack of matching sequences in the databases. These may also indicate the presence of new or novel species or members of cryptic species complexes. Though care was taken to match ITS2 to CO1 sequences towards the best possible identity, this was not possible in some cases where PCR amplification failed. This is the first time a comprehensive look at Culicinae mosquitoes has taken place at this site and points to the importance of saving non target species trapped as a by-product of a study. There was a high level of discrepancy between morphology and molecular identifications for Culicinae mosquitoes relative to Anophelinae mosquitoes. This may be attributed to a historically lesser focus on the development and validation of non-*Anopheles* morphological keys combined less research and consequently less experience on morphological identification related to these species. The presence of species complexes, sibling species and novel species in baseline characterizations (such as this) further complicates morphological identification. This is reflective of fewer studies (and consequent morphological identification experience) on non-anopheline mosquitoes in this region with a lack of ITS2 or CO1 sequences for some species, e.g. sequences for *Cx. sinensis* and *Ae. poicilius* are absent in the databases. This study identified *Cx. quinquefasciatus* as the dominant Culicinae species with highest relative abundance (determined morphologically). The role of *Cx. quinquefasciatus* as a vector of filaria was reported by other studies in Indonesia [32]. *Cx. tritaeniorhynchus*, *Cx. vishnui*, *Cx. pseudovishnui* and *Cx.fuscocephala*, identified in this study, tend to be indiscriminate feeders attracted to both human and animals [33]. The presence of the these *Culex* species combined with the local cultural practices of using domesticated pig as traditional currency point to a possible explanation for the maintenance of Japanese encephalitis transmission in the area [34].

Both *Armigeres malayi* and *Ar. subalbatus* were identified with peak biting during the crepuscular period with declining biting rates after, similar to that seen in other reports [35]. *Armigeres subalbatus* specifically demonstrated peak densities right after sunset, with no activity after 2100h –data supported by other studies that report similar temporal biting peaks [36,37].

*Anopheles* mosquitoes remain the genera with the highest identities with 100% of them having being morphologically and 24.55% (n = 7,276) being molecularly identified. A previous

study [38] trapped similar species with the exception of Group Hyrcanus species not being caught in this study. Furthermore, both *An. barbirostris* clade I *(An. barbirostris s.s.)* and *An. barbirostris* clade 2 *(An. vanderwulpi)* were confirmed to both be present at these sites along with *An. sundaicus s.l.* [39].

Temporal density of vectors overall were reliant on the availability of larval habitats driven by rainfall and agricultural irrigation [40]. The availability of multiple larval habitats throughout the year indicates a year round availability of nuisance mosquitoes as well as vectors of disease. Though *Culex* densities were the most impacted by rainfall, rainfall was a driver of other genera populations as well. Interestingly, the primary malaria vector along the coast, *An. sundaicus* [38,41,42] all but disappeared by December 2016 with a small recovery in 2017. Though the reasons for this decline is not yet clear, it may be attributed to contamination of larval sites (permanent and semi-permanent brackish water pools [42] by agricultural effluence rendering them hypoxic, or the community wide distribution of long lasting insecticide impregnated bed nets (LLINs) impacting adult populations [13].

Open house construction, typical of traditional Sumba houses, may contribute to the similar indoor and outdoor biting rates determined. Houses being mostly made of bamboo walls, elevated bamboo floors, with thatch roofs and open eaves allow for mosquito entry and exit. Data from Malawi demonstrate that an open house construction (similar to that seen in traditional Sumba houses) had higher (14x) malaria vector entry when compared to more closed houses [43]. Domestic animals, often kept under the elevated floors over the night, may also contribute to mosquitoes being attracted toward and into houses. Zoophagic mosquitoes such as *Armigeres* [44] may be attracted towards these animals under houses and consequently feed on humans instead.

The HBRs from night biting determined point to the highest biting rates in *Anopheles* and *Culex* mosquitoes with lower biting rates for *Aedes* and *Armigeres*. Interestingly, *Aedes* were found to bite, albeit at low levels, throughout the night. This finding is important as it demonstrates the potential of *Aedes*, based virus transmission in future studies that incorporate HLCs. Studies that characterize the diurnal biting profile of these mosquitoes would enable a 24 hour characterization of bionomic traits [35,36]. Equivalent indoor and outdoor biting rates indicate the need for interventions in both spaces as well as indicate that indoor interventions may also impact outdoor biting mosquitoes that may go indoors to feed [45,46].

This presence of multiple *Aedes* species indicate the potential transmission of *Aedes*-borne diseases such as Zika, dengue, chikungunya and Yellow fever; while the presence of *Armigeres* species point to transmission of Zika, filariasis and Japanese encephalitis [47–49]. Both *Anopheles* and *Culex* species had peak biting both indoors and outdoors before midnight reflecting human activity, with declining HBRs after. These periods represent temporal exposure where interventions may be most applicable towards reducing man vector contact. Multiple *Culex* species with relevant biting periods and rates may support the transmission of multiple *Culex*-borne diseases such as human filariasis and Japanese encephalitis that have been reported in the area [50]. The five most common *Anopheles* species reported here have been confirmed as vectors for malaria and filariasis [49]. Mosquitoes in the *Mansonia* genus were also found in this study, although at lower densities. *Mansonia* species have been documented in transmitting filariasis in Indonesia [51]. Nocturnal *B. malayi* is transmitted by *An. barbirostris*, which lives in paddy fields, and nocturnal sub-periodic *B. malayi* is transmitted by *Mansonia* species. In Sumba, two species of filarial parasite in human have been documented, *Wucheria bancrofti* and *B. timori* [52].

This study revealed a wide diversity of night-time human host seeking mosquito species in Sumba that potentially transmit multiple vector borne diseases. The presence of night biting vectors of non-malaria diseases warrants the careful application of the HLC technique to

ensure the safety of volunteers conducting the mosquito trappings [53,54], or the implementation of other innovative tools for adult mosquito collection, such as the double net trap [55] and Host Decoy Trap [44].

In conclusion, this study describes the array of human biting mosquitoes on the island of Sumba, Indonesia, demonstrating preliminary bionomic traits that impact human mosquito contact, and points to the potential of mosquito borne diseases that may be transmitted. This set of data is important for devising evidence-based vector borne disease mitigation strategies in Sumba and reveals the complexity of the mosquito biome in a single geography.

## Acknowledgments

The authors are grateful for the supports of the study volunteers in Southwest and West Sumba Districts, East Nusa Tenggara Province. The authors are grateful for the support of the Eijkman Institute for Molecular Biology (EIMB) Jakarta, Ministry of Health Republic of Indonesia, District health departments of Southwest and West Sumba, and East Nusa Tenggara Province. We appreciate the contribution of the entomology teams, local field workers, data entry clerks, and the numerous local volunteers for their dedication and active participation in this study. We thank Nadha Rizky Pratama, Sylvia Sance Marantina, Jenifer Kiem Aviani, and Annisa Rizkia for their assistance in the EIMB laboratory. This publication is dedicated to the memory of Dr. Michael Bangs who had a special interest in this site and study—where he imparted much experience and knowledge. Mike's contribution to the fight against malaria and other diseases across the world is prodigious and far reaching. He is greatly missed.

## Author Contributions

**Conceptualization:** Lepa Syahrani, Din Syafruddin, Puji B. S. Asih, Nicole L. Achee, Neil F. Lobo.

**Data curation:** Lepa Syahrani, Dendi H. Permana, Din Syafruddin, Siti Zubaidah, Puji B. S. Asih, Ismail E. Rozi, Anggi P. N. Hidayati, Sully Kosasih, Farahana K. Dewayanti, Nia Rachmawati, Rifqi Risandi, Michael J. Bangs, Claus Bøgh, Jenna Davidson, Allison Hendershot, Timothy Burton, John P. Grieco, Nicole L. Achee, Neil F. Lobo.

**Formal analysis:** Lepa Syahrani, Dendi H. Permana, Din Syafruddin, Siti Zubaidah, Puji B. S. Asih, Ismail E. Rozi, Nicole L. Achee, Neil F. Lobo.

**Funding acquisition:** Din Syafruddin, Nicole L. Achee, Neil F. Lobo.

**Investigation:** Lepa Syahrani, Din Syafruddin, Siti Zubaidah, Puji B. S. Asih, Ismail E. Rozi, Nicole L. Achee, Neil F. Lobo.

**Methodology:** Lepa Syahrani, Dendi H. Permana, Din Syafruddin, Siti Zubaidah, Puji B. S. Asih, Ismail E. Rozi, Anggi P. N. Hidayati, Sully Kosasih, Farahana K. Dewayanti, Nia Rachmawati, Rifqi Risandi, Michael J. Bangs, Claus Bøgh, Allison Hendershot, Timothy Burton, John P. Grieco, Nicole L. Achee, Neil F. Lobo.

**Project administration:** Din Syafruddin, Nicole L. Achee, Neil F. Lobo.

**Resources:** Lepa Syahrani, Dendi H. Permana, Din Syafruddin, Siti Zubaidah, Puji B. S. Asih, Nicole L. Achee, Neil F. Lobo.

**Software:** Lepa Syahrani, Dendi H. Permana, Ismail E. Rozi, Farahana K. Dewayanti, Nicole L. Achee, Neil F. Lobo.

**Supervision:** Din Syafruddin, Puji B. S. Asih, Michael J. Bangs, Nicole L. Achee, Neil F. Lobo.

**Validation:** Din Syafruddin, Puji B. S. Asih, Nicole L. Achee, Neil F. Lobo.

**Visualization:** Lepa Syahrani, Dendi H. Permana, Din Syafruddin, Puji B. S. Asih, Ismail E. Rozi.

**Writing – original draft:** Lepa Syahrani, Din Syafruddin, Neil F. Lobo.

**Writing – review & editing:** Lepa Syahrani, Dendi H. Permana, Din Syafruddin, Siti Zubaidah, Puji B. S. Asih, Ismail E. Rozi, Anggi P. N. Hidayati, Sully Kosasih, Farahana K. Dewayanti, Nia Rachmawati, Rifqi Risandi, Michael J. Bangs, Claus Bøgh, Jenna Davidson, Allison Hendershot, Timothy Burton, John P. Grieco, Nicole L. Achee, Neil F. Lobo.

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
