## [Decision Letter · Decision Letter 0]

3 Nov 2021

Dear Prof. Syafruddin,

Thank you very much for submitting your manuscript "An inventory of human biting mosquitoes and their bionomics in Sumba Barat, Indonesia" for consideration at PLOS Neglected Tropical Diseases. As with all papers reviewed by the journal, your manuscript was reviewed by members of the editorial board and by several independent reviewers. In light of the reviews (below this email), we would like to invite the resubmission of a significantly-revised version that takes into account the reviewers' comments. 

We cannot make any decision about publication until we have seen the revised manuscript and your response to the reviewers' comments. Your revised manuscript is also likely to be sent to reviewers for further evaluation.

Sincerely,

Uwem Friday Ekpo, PhD

Associate Editor

Wuelton Monteiro

Deputy Editor

Reviewer's Responses to Questions

**Key Review Criteria Required for Acceptance?**

**Methods**

-Are the objectives of the study clearly articulated with a clear testable hypothesis stated?

-Is the study design appropriate to address the stated objectives?

-Is the population clearly described and appropriate for the hypothesis being tested?

-Is the sample size sufficient to ensure adequate power to address the hypothesis being tested?

-Were correct statistical analysis used to support conclusions?

-Are there concerns about ethical or regulatory requirements being met?

Reviewer #1: The statistical analysis should be improved upon. The method used for human biting rate determination were not included in the methodology section.

Reviewer #2: (No Response)

**Results**

-Does the analysis presented match the analysis plan?

-Are the results clearly and completely presented?

-Are the figures (Tables, Images) of sufficient quality for clarity?

Reviewer #1: There is need to generate probably a line graph showing monthly human biting rates. Figures are not clear and of poor quality

Reviewer #2: (No Response)

**Conclusions**

-Are the conclusions supported by the data presented?

-Are the limitations of analysis clearly described?

-Do the authors discuss how these data can be helpful to advance our understanding of the topic under study?

-Is public health relevance addressed?

Reviewer #1: The conclusion is not adequate for the data presented. Major limitations were not clearly stated

Reviewer #2: (No Response)

**Editorial and Data Presentation Modifications?**

Reviewer #1: (No Response)

Reviewer #2: (No Response)

**Summary and General Comments**

Reviewer #1: This study documented an inventory of human biting mosquitoes and their bionomics in Sumba Barat, Indonesia using human landing catch between 18:00hr and 06:00 hr (i.e. 6:00pm to 6:00am). This study also try to relate abundance of the mosquito’s species to environmental factors such as temperature and rainfall.

This is a very interesting work, however it should be subjected to major revisions; 

General Comments;

1. This work was initially mainly focused on malaria transmitting mosquitoes, but other species were also trapped during the study collection. Hence the findings might not be the true situation in the study location, for instance, Aedes spp. bites more during the day and may even bite throughout the day. Please indicate this a major limitation to this study. If possible, change the title to capture this development

2. The conclusion in the abstract needs to be rewritten to buttress on the key findings. The conclusion looks like justification for the study.

3. The methodology needs restructuring. It should be broken down into the appropriate sections. For example, a lot of things have been lumped together under the section study site; study area, study design and part of the statement for statistical or data analysis section

4. The authors should avoid the indiscriminate use of hyphen as it makes most of the sentences not readable. Hyphen is better in separating words than sentences

5. The statistics employed was too basic for this kind of complex data. Hence the authors should see a statistician in order to improve the quality of the results and overall manuscript

6. If possible, the authors should try and dichotomize their mosquito data in a range, and run a mixed-effect-regression model to analyze the relationship between rainfall, temperature and abundance of different mosquito spp. 

7. No results were presented for monthly abundance alongside their entomological indices. You only presented general abundance. 

8. Your figures are not clear

9. The discussion needs improvement as results were repeated and a lot of statements were not clear

10. It seems this study was carried out by inexperienced ‘entomologist’ as stated by the authors. Hence one major co-founders for large discrepancies recorded between morphological and molecular identification. Were they properly trained for the identification procedures before embarking on the study?

Other Comments:

1. Line 89: Remove extra comma in Mansonia

2. Line 92: You can improve the readability of the statement by removing the hyphen and recast. E.g ………dry season owning to increased ………..

3. Line 102: Sentence is too short, merge the two sentences together to make one statement

4. Line 106: Remove the hyphen

5. Line 110-112: Statement is not clear, please recast.

6. Line 130: Over multiple years? Be concise, (April 2016: April 2018)

7. Line 142: Why spearman’s rho alone? Try and compute a linear regression model that will better quantify the magnitude of the relationships.

8. Line 157 – 159: Your morphological identification followed which identification guide? Please recast line 157 – 159 as they need to be reconciled. Did you identify the whole specimen or you did a random identification? Or the random ones selected were to confirm what have been previously identified? Please clarify

9. Under your analysis section, there are statements that are meant to be written under study design or protocol. You shouldn’t be stating here how you collected your data. Just mention the method of analysis and their importance to the variables used in this study.

10. The population of Aedes mosquitoes is low, could it be that it was under reported? Knowing well that it’s Aedes spp. can bite anytime of the day. However, I fully understood that this research was mainly focused on malaria transmitting mosquitoes according to your statements in some parts of your manuscript. It is therefore very important for you to properly state this a major limitation for your study.

11. Line 199: Are the order listed for the Culex, Aedes and Amigeres species in ascending or descending orders with respect to abundance? It will be good if you can put their percentage abundance in a brackets in front of each species

12. Line 209: Recast as The remaining 8 ………………… representing ___% of the total Anopheles comprised of An…………..

13. Line 216 – 219: Remove and relocate to the appropriate section under methodology

14. Line 235-236: Please remove hyphen and recast. E.g. however, the observed similarity was below………….

15. Line 201 – 252: Recast the statement and remove hyphen: e.g. Overall, there was an ------------Culicidae specimens (notably Culex and Anopheles) during the raining season

16. Line 254: P<0.48 is not correct for significant relationship. Rather put (r values ranging between 0.485 – 0.552, p<0.05).

17. How did you determine your human biting rates? This wasn’t spelt out in the methodology section

18. Line 322: fewer studies and lack of morphological identification experience? Do you have any literature to back it up? Are you saying the identifiers were not properly trained for morphological identification?

19. Line 327: Data that is supported? Statement not clear, please recast

20. Line 373: Remove figure 5. Also avoid stating your results in the discussion section

Reviewer #2: please see attachment for all comments

PLOS authors have the option to publish the peer review history of their article (what does this mean?). If published, this will include your full peer review and any attached files.

Reviewer #1: No

Reviewer #2: No
---

## [Decision Letter · Decision Letter 1]

9 Mar 2022

Dear Prof. Syafruddin,

We are pleased to inform you that your manuscript 'An inventory of human night-biting mosquitoes and their bionomics in Sumba, Indonesia' has been provisionally accepted for publication in PLOS Neglected Tropical Diseases.

Best regards,

Uwem Friday Ekpo, PhD

Associate Editor

Wuelton Monteiro

Deputy Editor

Reviewer's Responses to Questions

**Key Review Criteria Required for Acceptance?**

**Methods**

-Are the objectives of the study clearly articulated with a clear testable hypothesis stated?

-Is the study design appropriate to address the stated objectives?

-Is the population clearly described and appropriate for the hypothesis being tested?

-Is the sample size sufficient to ensure adequate power to address the hypothesis being tested?

-Were correct statistical analysis used to support conclusions?

-Are there concerns about ethical or regulatory requirements being met?

Reviewer #1: The study design is appropriate to address the stated objectives. Correct Statistics were done to support conclusions. No concern about ethical or regulatory requirements were observed.

Reviewer #2: (No Response)

**Results**

-Does the analysis presented match the analysis plan?

-Are the results clearly and completely presented?

-Are the figures (Tables, Images) of sufficient quality for clarity?

Reviewer #1: The analysis presented match the analysis plan. The results were clearly and completely presented. The figures and tables were reasonably clear

Reviewer #2: (No Response)

**Conclusions**

-Are the conclusions supported by the data presented?

-Are the limitations of analysis clearly described?

-Do the authors discuss how these data can be helpful to advance our understanding of the topic under study?

-Is public health relevance addressed?

Reviewer #1: The conclusions were supported by the data presented. The authors extensively discuss how the generated data can be helpful to advance our understanding of the topic under study. The public health relevance of the study was adequately addressed.

Reviewer #2: (No Response)

**Editorial and Data Presentation Modifications?**

Reviewer #1: (No Response)

Reviewer #2: (No Response)

**Summary and General Comments**

Reviewer #1: (No Response)

Reviewer #2: (No Response)

PLOS authors have the option to publish the peer review history of their article (what does this mean?). If published, this will include your full peer review and any attached files.

Reviewer #1: No

Reviewer #2: No

---

## [Editor Report · Acceptance letter]

18 Mar 2022

Dear Prof. Syafruddin,

We are delighted to inform you that your manuscript, "An inventory of human night-biting mosquitoes and their bionomics in Sumba, Indonesia," has been formally accepted for publication in PLOS Neglected Tropical Diseases.

Best regards,

Shaden Kamhawi

co-Editor-in-Chief

Paul Brindley

co-Editor-in-Chief
